# Utility of the Full ECG Waveform for Stress Classification

**DOI:** 10.3390/s22187034

**Published:** 2022-09-17

**Authors:** Katya Arquilla, Andrea K. Webb, Allison P. Anderson

**Affiliations:** 1Department of Aeronautics and Astronautics, Massachusetts Institute of Technology, Cambridge, MA 02139, USA; 2Draper, Cambridge, MA 02139, USA; 3Smead Aerospace Engineering Sciences, University of Colorado Boulder, Boulder, CO 80309, USA

**Keywords:** electrocardiography, feature extraction, heart rate variability, psychophysiology, stress

## Abstract

The detection of psychological stress using the electrocardiogram (ECG) signal is most commonly based on the detection of the R peak—the most prominent part of the ECG waveform—and the heart rate variability (HRV) measurements derived from it. For stress detection algorithms focused on short-duration time windows, there is potential benefit in including HRV features derived from the detection of smaller peaks within the ECG waveform: the P, Q, S, and T waves. However, the potential drawback of using these small peaks is their smaller magnitude and subsequent susceptibility to noise, making them more difficult to reliably detect. In this work, we demonstrate the potential benefits of including smaller waves within binary stress classification using a pre-existing data set of ECG recordings from 57 participants (aged 18–40) with a self-reported fear of spiders during exposure to videos of spiders. We also present an analysis of the performance of an automated peak detection algorithm and the reliability of detection for each of the smaller parts of the ECG waveform. We compared two models, one with only R peak features and one with small peak features. They were similar in precision, recall, F1, area under ROC curve (AUC), and accuracy, with the greatest differences less than the standard deviations of each metric. There was a significant difference in the Akaike Information Criterion (AIC), which represented the information loss of the model. The inclusion of novel small peak features made the model $4.29\times 10^{28}$ times more probable to minimize the information loss, and the small peak features showed higher regression coefficients than the R peak features, indicating a stronger relationship with acute psychological stress. This difference and further analysis of the novel features suggest that small peak intervals could be indicative of independent processes within the heart, reflecting a psychophysiological response to stress that has not yet been leveraged in stress detection algorithms.

## 1. Introduction

Psychological stress causes changes in the autonomic nervous system that can be observed in the electrocardiogram (ECG) signal. The R peak—the most prominent feature in the ECG signal, shown in Figure 1—is most commonly used to derive features in the time domain for the classification of psychological stress because it is relatively easy to detect under noisy conditions and requires less infrastructure to collect (e.g., the R peak can be detected with fewer electrodes and in lower signal-to-noise ratio signals than the smaller peaks in the waveform). Features in the time domain also tend to provide more reliable metrics over short-duration monitoring periods than features extracted from the frequency domain. This presents the question of whether there are more features available within the time domain of the ECG signal beyond traditional heart rate variability (HRV) features derived only from the R peak and whether these novel features (derived from the P, Q, S, and T peaks) can be leveraged to improve stress classification algorithms. This exploration is supported by the advancement of many wearable ECG detection systems that allow the collection of the full ECG waveform, driving the possibility of the operational utility of these novel features and a clear paradigm for their implementation.

Many studies have explored the utility of the ECG signal for psychological stress detection and have shown that traditional R-peak-based HRV features can reliably indicate stressed state [1,2]. To the best of our knowledge, these studies have not explored the efficacy of small peak features, largely due to the fact that the small peaks have not been easily detectable without research-grade ECG recording equipment. Other studies have pursued a multimodal approach to both physiological and psychological stress classification with the inclusion of other signal streams—electrodermal activity (EDA) being the most common—in their feature sets, because each additional signal stream provides some reduction in collinearity due to the difference in physiological processes being measured [3,4,5,6,7]. In this investigation, we strive to understand whether the incorporation of small peak features can bring the same sort of reduction in collinearity to stress classification because of the subtle physiological differences between the production of the R peak and the smaller peaks within the heart. These parts of the heart beat cycle are less well-studied in psychophysiology, so in this work we hope to add to our understanding of their psychophysiological importance and their potential utility for stress classification. The full set of potential small peak intervals available to our classification algorithm before a feature down-select are shown in Figure 1.

This more scientific exploration of small peak utility must be accompanied by an understanding of the operational utility of these features. As previously mentioned, the development of wearable ECG-monitoring systems is increasing rapidly, but wearable systems are plagued with noise due to poor electrode contact and motion artifacts [8]. For this reason, we also aimed to understand the effects of noise on automated peak detection for each of the five peaks within the waveform to identify which small peaks would be lost first under noisy conditions.

The research questions we address in this work are threefold: (1) Do features derived from the intervals between the P, Q, S, and T waves within the ECG waveform augment binary stress classification? (2) Can we characterize the correlations between the novel and traditional features to improve our understanding of the influence of the novel features? (3) What is the impact of noise on the detectability of these smaller parts of the ECG waveform? Our goal is to understand both the efficacy of adding these features to stress detection algorithms and the practicality of their use in operational environments where noise is ubiquitous.

The paper is organized as follows: Section 2 describes the methods of model development, pre-existing data set used for the analysis, model comparison, and peak detection performance, Section 3 presents the features included in the traditional feature model (TFM) and novel feature model (NFM), the comparisons between the two, and the results of the peak detection performance analysis. Lastly, in Section 4, we situate our results within the gaps in understanding in this area and cover the limitations of this study.

## 2. Model Development Methods

Each of the stress classification models was developed by down-selecting time-domain ECG features that were extracted from a pre-existing data set, assessing validity coefficients for each feature, inspecting correlations between the features, and ultimately assessing model performance with three widely used classification algorithms. Outside of model development, an assessment of automated peak detection performance was also conducted to understand the potential operational utility of small peak features. The TFM and NFM were both developed using the set of traditional features, but the NFM was also given the full set of potential novel features before the model down-selection, while the TFM was not. The NFM was not based purely on small peak features but instead on a combination of traditional and novel features. The following sections step through these development procedures in more detail for both the TFM and NFM and the methods used to evaluate their performance.

### 2.1. Data Set

The data set used for this study is a data set publicly available on PhysioNet, a database of physiological signal data sets managed by the National Institutes of Health [9]. This data set was created as part of an investigation of exposure therapy as a method of mitigating arachnophobia in spider-fearful individuals and was downloaded for this study in July 2020. The data set contains ECG recordings from 57 participants (aged 18–40) with reported fear of spiders while they were exposed to 16 different video clips of spiders in the wild with short rest periods in between [10]. We acknowledge here that fear and stress are not the same psychological construct, but their physiological manifestations are similar, and we will revisit the impact of this limitation in Section 4. ECG data collection was conducted using a wearable BITalino biosignal measurement device, and the study was approved by the Ethical Committee of the Faculty of Human Sciences of Saarland University, as stated by the authors of the original study. Each ECG recording is accompanied by a time-stamp log of video delivery, and we used these stamps to label each time stamp as unstressed —0—or stressed —1— and used these labels as the true values for classification. This approach was taken due to the lack of the subjective self-assessments of participants’ responses to the video stimuli. Therefore, we cannot state that these classifications are absolutely true, but they provide a valuable proxy for stressed state. In this analytic structure, there are 16 ten-second samples each for stressed and nonstressed states from each of the 57 participants. This totals 1824 samples for model training and testing, but only those samples where both the R peak and small peak features could be reliably detected were used, resulting in 960 samples left for model training and testing. All data processing was performed using the publicly available Python package neurokit2 [11].

### 2.2. Signal Denoising and Feature Extraction

Each ECG recording was filtered using a 5^*th*^- order Butterworth filter with a normalized cutoff frequency of 0.3 and subsequently split into 10 s samples, one from each stressed and nonstressed period in the data set. We selected sample durations of 10 s to accommodate the shortest intervals between video clips. These short time intervals were also relevant for psychological stress detection in real time. Peak detection was conducted using the “findpeaks” and “delineate” functions based on the Pan–Tompkins algorithm for peak detection available in the neurokit2 Python package [12,13]. Novel features consist of the mean and standard deviation of time intervals between the P, Q, R, S, and T peaks and were extracted by subtracting the time stamps between the selected peaks. After feature extraction, each feature was converted to a z-score to scale and center the data, calculated across participants. Only time domain features were used for this study because the 10 s sample lengths were too short to derive reliable and meaningful frequency domain features.

#### Explanation of Features

This section provides an explanation of each of the features included in both models. There were a total of 24 R-peak-based features and a total of 12 small-peak-based features for each of the models to down-select from. In this paper, we have only included the features that were ultimately selected by the two models in this explanation. Each feature is described in Table 1.

### 2.3. Model Development Procedure

The same method of feature selection was used for both the TFM and the NFM. We split the 960 samples into a training set of 67% and a testing set of 33%. The full set of available features for both models was reduced using a forward and backward stepwise feature selection algorithm with the Akaike Information Criterion (AIC) as the evaluation metric. The smaller the AIC value, the less information loss in the model, so at each step the feature with the highest AIC value was removed from the set.

Validity coefficients were computed for each feature by calculating the point-biserial correlation, $r_{pb}$, between the criterion (i.e., stress or no stress) and the feature values as an additional check of the results of the stepwise feature selection procedure. In addition to these validity coefficients, feature distributions were examined in a histogram visualization to observe the differences between the feature distributions for stressed and nonstressed samples through visual inspection. After the feature down-selection for both models, Akaike Information Criterion (AIC) scores were compared between the two to measure the difference in information loss between the models.

### 2.4. Comparison of Multiple Classification Algorithms

We evaluated three different classification methods with both the TFM and NFM. These three methods were linear discriminant analysis (LDA), logistic regression (LR), and a support vector classification (SVC). These formulations are different in their approaches, and the goal with this investigation was to see how well the two feature sets performed under different, but commonly used, classification algorithms. LDA is based on least squares regression, whereas LR is based on maximum likelihood estimation. LDA assumes normality and homoscedasticity in the data set under the two conditions (i.e., stressed and not stressed), while LR does not. SVC is a maximum margin classifier that maximizes the distance between each feature and the hyperplane decision boundary. Explainability is highest in the LDA and LR models and lowest in the SVC model. We compared the performance of these three classification algorithms in precision, recall, F1, AUC, and accuracy with both the TFM and NFM. Each of these metrics is presented as an average value from 10-fold cross-validation conducted on the reserved test data.

### 2.5. Peak Detection Algorithm Performance Analysis

If the small peak features show promise for stress detection, the capabilities of automated peak detection algorithms must be evaluated to ensure operational feasibility. It is time-consuming to identify features through visual inspection, so the smaller parts of the ECG waveform must be detectable by unsupervised peak detection algorithms in order to be effectively incorporated into stress detection algorithms that function without constant supervision. To assess the detectability of each feature, we randomly selected 10 of the 57 participants’ data sets to assess precision, recall, and F1 scores for the neurokit2 peak detection function when compared with peak detection through visual inspection. The peak detection algorithm begins by calculating the gradient of the ECG signal to identify the location of the QRS complexes within the signal and identifies each of the peaks by calculating local maxima within that region.

## 3. Results

### 3.1. Traditional Feature Model

The TFM consists of nine features down-selected from the 24 traditional features derived from R-R interval values. The features present in this model are RMSSD, MeanNN, CVSD, MedianNN, IQRNN, pNN50, pNN20, TINN, and HTI. In this set of features, CVSD has the highest coefficient in the model, and MeanNN and MedianNN have the strongest correlation with stress. Each of these metrics displays a negative correlation with stress, meaning that lower values for each feature are associated with greater stress. The results are summarized in Table 2.

### 3.2. Novel Feature Model

The NFM consists of 13 features down-selected from the 36 features drawn from R-R-interval-based HRV metrics and small-peak-based metrics. The features present in this model are SDNN, MadNN, MCVNN, IQRNN, pNN50, pNN20, TINN, PR mean, PR sd, ST mean, ST sd, PT mean, and PS sd. Four of the traditional HRV features overlap with those selected in the TFM: IQRNN, pNN50, pNN20, and TINN. None of the novel features derived from the Q peak contributed positively to the AIC metric and none were selected as part of the final NFM. As with the features in the TFM, the features included in the NFM are negatively correlated with stress level. These results are shown side-by-side with the TFM results in Table 2.

### 3.3. Multicollinearity and Validity Coefficient Assessments

The top of Figure 2 shows the correlation matrix for the features included in the TFM. RMSSD is strongly correlated with four of the other features in the model (CVSD, IQRNN, pNN50, and TINN). Each of these four features is also strongly correlated with the other members of the group, including RMSSD. The only feature that shows a small negative correlation with all the other features is HTI, which is likely due to the fact that HTI is a geometric measure of the R-R interval distribution, unlike the other features. The traditional features show higher correlation coefficients amongst themselves than they show with the novel features. The novel features show lower correlation coefficients with each other and with the traditional features, making them potentially more independent and informative than just the R-peak-based traditional features. The HTI feature is not shown in this figure to improve interpretability because it is positively correlated with stress (opposite to the rest of the features).

The bottom of Figure 2 shows the correlation map for the features included in the NFM. The strongest correlations between the small peak features and traditional HRV features are between the mean PR, ST, and PT intervals and the MedianNN feature. The small peak standard deviations in general show lower correlation with the mean small peak interval features.

In addition to inspecting the correlation coefficients between the features, we calculated the validity coefficients between each feature and stress. The coefficients for each feature are shown in Table 2, and Figure 3 shows an example of the histograms we inspected for each feature that show the difference in distributions between stressed and nonstressed state. The left histogram shows the difference for the PT mean feature, which has the strongest validity coefficient, and the bottom histogram shows MCVNN, which has the weakest correlation. The histogram data representation shows the difference in validity correlation between the features, with the PT mean showing a clear difference in centering and spread between the two states and the MCVNN showing very similar distributions for both. The distribution of the PT mean feature values is both centered at a lower value and tighter in the stressed samples than in the unstressed samples. PT mean has a high validity coefficient, matching this visual representation. The two distributions for the MCVNN feature are similar in shape and location, reflecting the lower validity coefficient of this feature. This illustrates the greater predictive power of the novel PT mean feature as compared to the traditional MCVNN feature.

### 3.4. Model Comparison

Table 3 shows the features present in the TFM and NFM with their standardized regression coefficients (β), p-values, and validity coefficients ($r_{pb}$). The β coefficients are standardized regression coefficients without units, so their values signify the number of standard deviations the dependent variable will change with a change of one standard deviation in the corresponding feature. The β metric for the CVSD feature in the TFM is higher than for all the other features, and the β metrics for all six of the novel features and the MCVNN traditional feature are higher than for the rest of the features in the model. MeanNN, MedianNN, PR mean, ST mean, and PT mean have the highest validity coefficients across the set of features in both models. AIC values were calculated using the test data for both models. The AIC value for the TFM is −2105.24, and the AIC value for the NFM is −2237.1. These scores must be compared by calculating their difference as exponentials [14]. This calculation shows that the TFM is $2.33\times 10^{-29}$ times as probable as the NFM to minimize the information loss, showing that the NFM is more likely to preserve information than its counterpart.

### 3.5. Peak Detection Algorithm Performance

The distributions of precision, recall, and F1 scores for each part of the ECG waveform are shown in Figure 4. Precision indicates the correctly identified peaks, or “hits”, and recall indicates the portion of peaks that were correctly identified, or the “hit rate”. As shown in Table 4, the P peak and Q peak both show decrements in precision compared to the other peaks. The Q peak is most often misidentified. The Q peak is the most difficult to detect when noise is present, as is shown in the distribution of recall values. In the data where the Q peak is most difficult for the algorithm to detect correctly, the Q peak does not create a large deviation from the beginning of the R peak. Automated Q peak detection produces high false positives and misses, resulting in relatively low precision, recall, and F1 scores. However, the standard deviation of each of these metrics across samples is high, and this is reflected in the non-normal distribution of scores for the Q peak in recall and F1. Some participants’ data show strong Q peaks, while others do not. In this analysis, motion artifacts that produce large-scale features within the data are most commonly incorrectly identified as ectopic beats, and the false detection of an R peak leads to the subsequent false detection of the smaller peaks.

## 4. Discussion

The results of this study suggest that the inclusion of small peak features in stress classification algorithms can reduce information loss, potentially increasing the predictive power of these algorithms, even with short ECG samples. We can see this result in the comparison of the TFM with only R-peak-based features and the NFM with both R-peak- and small-peak-based features.

The NFM performs slightly better than the TFM in the commonly used precision, recall, F1, AUC, and accuracy metrics, but they are not statistically significantly improved, given that the differences are within the standard deviations of each value. The differences between the two models in these metrics are largest with LR classification, and they perform almost identically with the SVC. These results do not show conclusively that the inclusion of small peak features augments the binary classification in this case, but they do suggest that these features can be used effectively in conjunction with traditional HRV features. However, in the AIC metric comparison, the NFM does perform better than the TFM. The NFM has a substantially better AIC score than the TFM, showing that the NFM is $4.29\times 10^{28}$ times more probable to minimize the information loss than the TFM. This difference could be due to the reduction in multicollinearity brought about by the incorporation of small peak features or the higher validity coefficients between the small peak features and stress. The AIC is based on the Kullback–Liebler divergence between the predicted value distribution and the true value distribution, so it measures information loss (i.e., the balance between the model’s predictive power and its simplicity) [15]. This differs from the other comparison metrics shown in Table 3 because they measure only predictive power. This could explain the difference in the AIC scores between the models.

Mayya et al., 2015, explored the efficacy of time domain, frequency domain, and nonlinear heart rate variability (HRV) features over 60 s test intervals and found that RMSSD was the most influential feature for determining stressed state when model performance was measured using classification accuracy [16]. In our investigation, the TFM includes this feature, but the RMSSD feature is not included in the NFM. This suggests that the importance of this feature is eclipsed by the presence of the small peak features that are more strongly correlated with stressed state over the short sample length (10 s) used in this study.

The β coefficients of the small peak features are all higher than those of the traditional features. Table 2 shows their values ranging from changes of 1.78 to 6.78 standard deviations in the response variable with a 1 standard deviation change in each feature. This indicates that over this short time-scale (10 s), the small peak features have far more predictive power for psychological stress detection than their traditional counterparts.

This, the difference in information loss between the two models, the reduction in multicollinearity, higher validity coefficients, and the inclusion of small peak features over other commonly included features, suggest that the small peak intervals could be indicative of somewhat independent processes within the heart that reflect the reaction of the autonomic nervous system to stress as seen in the ECG signal.

The novel features included in the NFM were all derived from the P, S, and T peaks. These peaks reflect known physiological processes, but they have not been studied deeply as psychophysiological correlates of stress. The duration of the P-R interval is dependent upon signal conduction through the atria, atrioventricular node, His bundle, and Purkinje fibers [17]. This conduction velocity determines the length of time allowed for the passage of blood from the atria to the ventricles. While we do not know the relationship between this difference and stress, our results show that the P-R interval mean and standard deviation are both negatively correlated with stress (e.g., shorter intervals and a tighter distribution of intervals are correlated with stress). The S-T interval falls within the period of ventricular repolarization, the relaxation and preparation for the next heartbeat [18]. Some studies have shown that the S-T interval is correlated with blood pressure; in our study, the shorter S-T interval mean and smaller standard deviation were found to be correlated with stress, and we hypothesize that this may be due to a correlation between blood pressure and stress [19]. Work by Pollak and Obrist suggests that pulse transit time (PTT) may be measurable through time intervals within the ECG signal [20]. PTT is usually measured with the ECG signal and a secondary pulse measurement, so measuring this within the ECG signal itself could reduce the number of sensors required for this psychophysiologically relevant signal to be collected, simplifying operational monitoring systems. While these are all potential explanations for the importance of these features in the NFM, we cannot make any definitive statements from this study. Further testing with multimodal sensor systems should be conducted to understand this relationship fully.

From an operational standpoint, the S and T peaks are most reliably detected with an automated detection algorithm, making them the most reliable for inclusion in stress classification models. The Q peak is least reliably detected, due to its smaller magnitude in comparison with the other peaks. The intervals associated with the Q peak were not included in the NFM, so this result does not negatively impact the potential for operational use of the NFM.

In this investigation, we made the assumption that exposing spider-fearful individuals to videos of spiders would create a psychophysiological response shown in the ECG signal similar to responses indicative of psychological stress. In the study conducted by Ihmig et al., HRV metrics were extracted for their correlation with “disturbance and emotional arousal”, so we believe that our assumption is valid [21]. Additionally, it has also been shown in other studies that both fear and stress are related to arousal and increases in heart rate [22].

Future work may include the validation of the TFM and NFM on other data sets. Testing with other data sets involving other types of psychological stress (e.g., social stress) could add to our depth of understanding in the utility of small peak features in this paradigm. Additionally, we plan to investigate the utility of small peaks for nonbinary stress classification, determining whether their inclusion augments the detection of multiple levels of stress. Another important area of future work is the validation of the NFM with different types of stress (e.g., workload stress, emotional stress, etc.) and the investigation of whether small peaks contribute to the classification of different types of stress.

In parallel with further assessments of the utility of the NFM in different experimental scenarios with nonbinary stress classification, an investigation of novel peak detection algorithms could expand the potential for use in the NFM. There are several groups working to develop peak detection algorithms that amplify the small peaks within the ECG waveform, and leveraging their work in this area is an important next step. In the future, it may be valuable to conduct a study of the efficacy of adding small peaks in conjunction with the use of such an algorithm.

## 5. Conclusions

This investigation focused on the benefit and utility of incorporating features derived from small peaks within the ECG waveform for binary stress classification. To the best of our knowledge, these features have not been investigated in depth for stress detection, so we hope the results of this study contribute renewed interest in their investigation as wearable sensor systems develop and become more capable of detecting small ECG peaks reliably. We found that the NFM performed slightly better than the TFM in the commonly used metrics of precision, recall, F1, AUC, and accuracy, but not statistically significantly so. Alternatively, we found that the NFM exhibited a better AIC score than the TFM, indicating that the NFM has less information loss than the TFM. This difference could be due to the higher validity coefficients between the small peak features and most of the traditional HRV features and the lower correlations between these features, potentially reducing multicollinearity within the model. These results show that the incorporation of these features into future psychological stress detection algorithms could augment these algorithms’ capabilities. As to the question of operational feasibility, we found that the Q peak was least robust to noise, so its use is not feasible for stress detection outside of the laboratory environment; however, the Q peak was not included in the NFM, so it does not show promise for stress detection and its poor detectability is not of major concern.

## Figures and Tables

**Figure 1 sensors-22-07034-f001:**
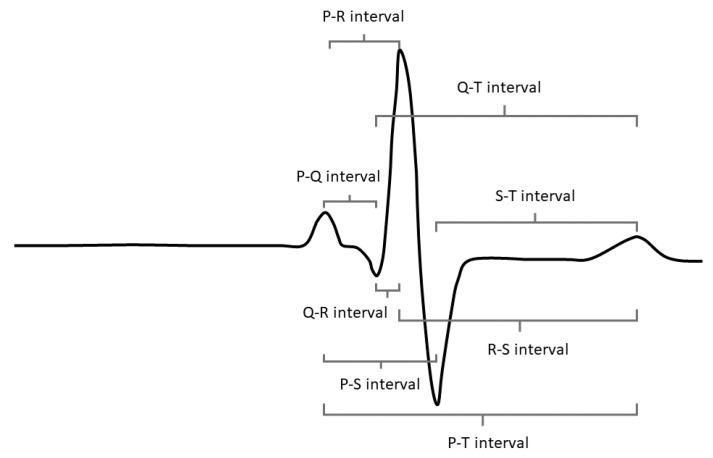
All waveform features of a classic ECG signal. This includes small peak intervals entered as candidate features into the novel feature model (NFM).

**Figure 2 sensors-22-07034-f002:**
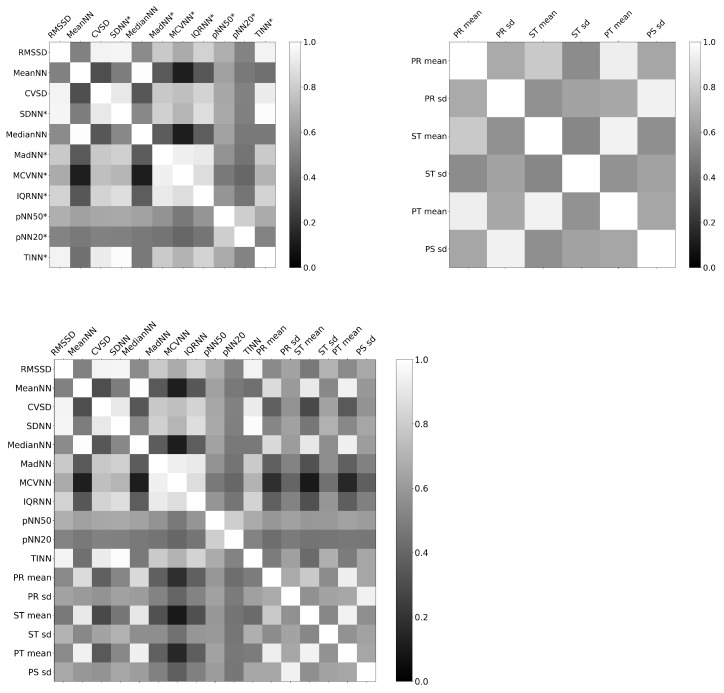
Correlation matrices for features included in the TFM and NFM. The top left matrix shows Pearson correlation coefficients between R-peak-derived traditional features; the features with an asterisk (*) are included in both the TFM and NFM, and those without are only included in the TFM. The top right matrix shows the correlation coefficients between the novel small-peak-derived features that are only included in the NFM. The bottom matrix shows all features in both models.

**Figure 3 sensors-22-07034-f003:**
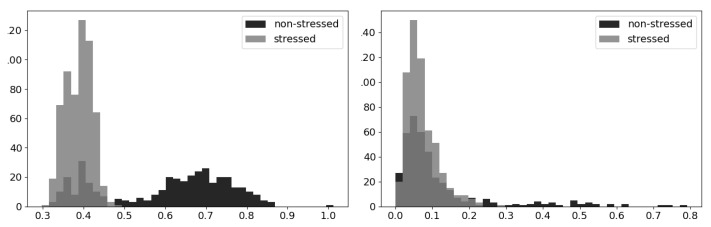
These histograms provide a way to visualize the relationship between each feature and stress and the related validity coefficients. **Left**: Distributions of the PT mean feature for stressed and nonstressed state; **right**: distributions of the MCVNN feature for the same two states.

**Figure 4 sensors-22-07034-f004:**
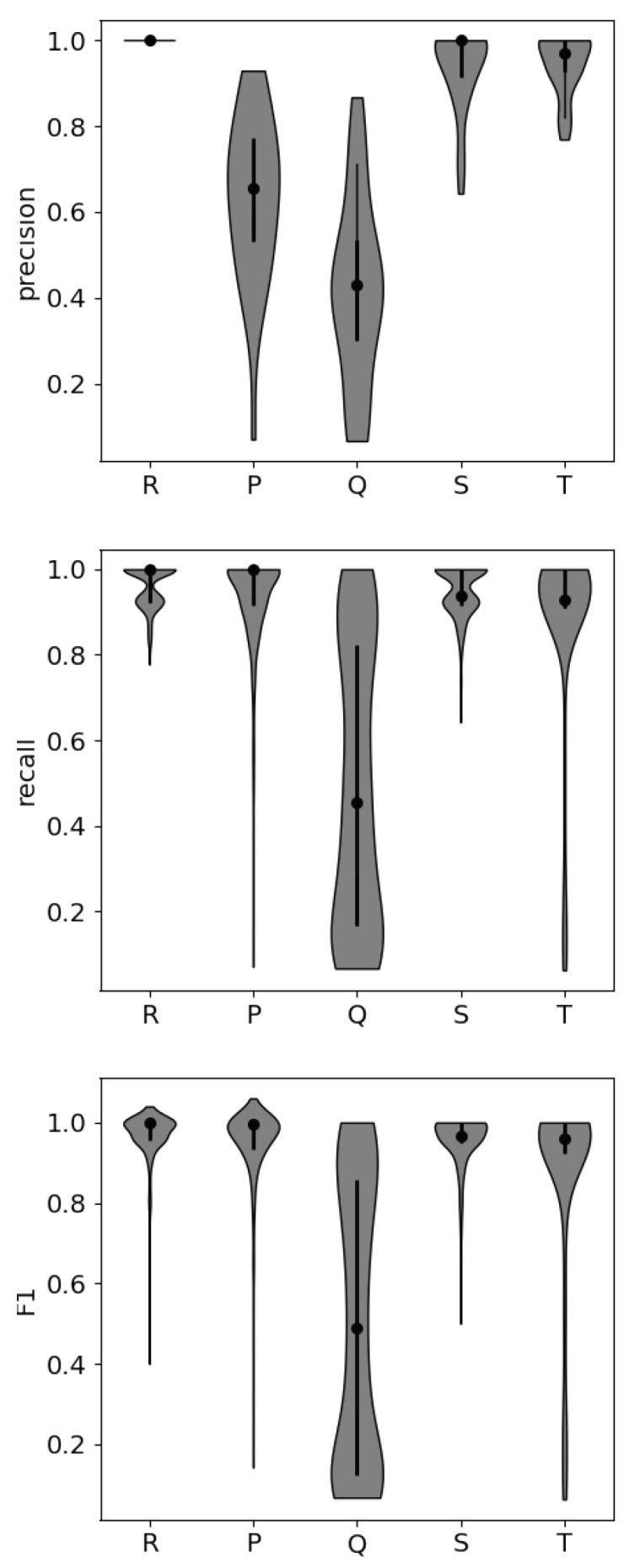
Precision, recall, and F1 scores for each peak within the ECG waveform from 10 participants selected at random. Each participant’s data comprise 32 different 10 s samples, half stressed and half unstressed. The shaded gray area shows the distribution of scores, the black dot shows the median value, and the thick black central line shows the interquartile range. The detection of the Q peak shows the lowest median scores and highest degree of variability between samples, making its detection unreliable.

**Table 1 sensors-22-07034-t001:** List of heart rate variability (HRV) features and their meaning. The right column states whether they are traditional (T) or novel (N) features.

Feature	Explanation	T/N
RMSSD	root mean square of the successive differences between R-R intervals	T
MeanNN	mean R-R interval	T
SDNN	standard deviation of R-R intervals	T
MadNN	median absolute deviation of R-R intervals	T
MCVNN	median-based coefficient of variation	T
CVSD	coefficient of the variation of successive differences	T
MedianNN	median of the absolute values of the successive differences between R-R intervals	T
IQRNN	interquartile range of R-R intervals	T
pNN50	percent of R-R intervals greater than 50 ms	T
pNN20	percent of R-R intervals greater than 20 ms	T
TINN	baseline width of the distribution of R-R intervals	T
HTI	the total number of R-R intervals divided by the height of the histogram of R-R intervals	T
PR mean	mean P-R interval	N
PR sd	standard deviation of P-R interval	N
ST mean	mean S-T interval	N
ST sd	standard deviation of S-T interval	N
PT mean	mean P-T interval	N
PS sd	standard deviation of P-T interval	N

**Table 2 sensors-22-07034-t002:** Coefficients, *p*-values, and validity coefficients for each feature as they are included in both the traditional and novel models. All validity coefficient values had *p*-values < 0.01. The TFM includes 9 traditional R peak features, and the NFM includes 7 traditional R peak features and 6 novel small peak features for a total of 13 features. There is overlap between the two models in four of the traditional features: IQRNN, pNN50, pNN20, and TINN.

	Traditional		Novel		
**Feature**	β	* **p** * **-Value**	β	* **p** * **-Value**	$r_{\mathrm{pb}}$
RMSSD	0.004	<0.01			−0.49
MeanNN	−0.001	<0.01			−0.70
SDNN			0.004	<0.01	−0.48
MadNN			0.001	<0.01	−0.38
MCVNN			−1.27	0.01	−0.22
CVSD	−2.37	<0.01			−0.36
MedianNN	−0.001	0.04			−0.70
IQRNN	−0.0004	<0.01	−0.001	<0.01	−0.39
pNN50	0.002	0.02	0.003	<0.01	−0.45
pNN20	0.004	<0.01	0.003	<0.01	−0.29
TINN	−0.0002	0.122	−0.001	<0.01	−0.44
HTI	−0.031	0.109			0.28
PR mean			4.84	<0.01	−0.70
PR sd			−3.52	0.03	−0.53
ST mean			5.40	<0.01	−0.66
ST sd			−1.78	<0.01	−0.52
PT mean			−6.78	<0.01	−0.73
PS sd			3.51	0.01	−0.52

**Table 3 sensors-22-07034-t003:** Performance metrics for the TFM and NFM with three different classification models. Each value is the mean of 10-fold cross-validation plus or minus the standard deviation. The values in bold font are higher than their counterparts in the other model.

LDA	Traditional	Novel
Precision	0.86 ± 0.06	**0.87 ± 0.06**
Recall	1.00 ± 0.01	1.00 ± 0.00
F1	0.92 ± 0.03	**0.93 ± 0.04**
AUC	0.89 ± 0.09	**0.91 ± 0.07**
Accuracy	0.89 ± 0.05	**0.91 ± 0.05**
**Logistic Regression**	**Traditional**	**Novel**
Precision	0.84 ± 0.06	**0.86 ± 0.06**
Recall	0.94 ± 0.07	**0.96 ± 0.04**
F1	0.89 ± 0.05	**0.91 ± 0.04**
AUC	0.87 ± 0.08	**0.90 ± 0.08**
Accuracy	0.85 ± 0.06	**0.87 ± 0.06**
**SVC**	**Traditional**	**Novel**
Precision	0.86 ± 0.06	0.86 ± 0.06
Recall	1.00 ± 0.00	1.00 ± 0.00
F1	**0.93 ± 0.03**	0.92 ± 0.03
AUC	0.89 ± 0.08	0.89 ± 0.08
Accuracy	0.90 ± 0.05	0.90 ± 0.05

**Table 4 sensors-22-07034-t004:** Precision, recall, and F1 metrics for automated peak detection of each of the parts of the ECG waveform. These results were derived from the visual inspection of 10 randomly selected participants’ data. The Q peak shows the most vulnerability to noise, with the lowest precision, recall, and F1 values and the largest standard deviations in all three.

Peak	Precision	Recall	F1
R	0.98 ± 0.07	0.96 ± 0.05	0.97 ± 0.06
P	0.94 ± 0.14	0.93 ± 0.11	0.95 ± 0.10
Q	0.48 ± 0.35	0.49 ± 0.33	0.50 ± 0.35
S	0.96 ± 0.09	0.95 ± 0.06	0.95 ± 0.07
T	0.88 ± 0.27	0.87 ± 0.24	0.86 ± 0.25

## Data Availability

The publicly available data set can be found here: https://physionet.org/content/ecg-spider-clip/1.0.0/, accessed on 10 July 2022. The code developed by the authors to analyze the data set can be found here: https://github.com/ksarquilla/stress_detection_ECG, accessed on 10 July 2022.

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
