# Peer review of "Utility of the Full ECG Waveform for Stress Classification"

_sensors, 2022, doi:10.3390/s22187034_

Round 1
Reviewer 1 Report
1. The overall scientific writing is not satisfactory enough. It reads more like a student's lab report than a journal article. Thorough revision is necessary before resubmission.
2. P1L4, what is Draper?
3. Abstract, insufficient information on method, dataset description and quantitative results. Please add such kind of information briefly.
4. Abstract, “The detection of psychological stress using … difficult to reliably detect.” is not necessary. It is better to be deleted or moved to Introduction section.
5. Fig. 1, duplicate “R-S interval”. Please delete one.
6. P2L70, Section ??, please confirm and revise.
7. Why two datasets were used?
8. How to annotate the “un-stressed” and “stressed”? What is the ground-truth for your annotation?
9. Section 2.2.1 and others, please tabulate such contexts.
10. Fig. 2, Incomplete keywords
11. Fig. 2, top, middle, bottom -> top left, top right, bottom
12. Tables, the title should go to the top of a table.
13. Using small peaks is a good idea. However, detection of small peaks is not easy and reliable enough usually. Please discuss this issue and the impact on your classification performance.
14. Please enhance the psychophysiological background and principle for including small peaks as new features.
Author Response
Reviewer 1:
- P1L4, what is Draper?
The location information for Draper, a not-for-profit laboratory, has been added in the information for the second author. This should address this question.
- Abstract, insufficient information on method, dataset description and quantitative results. Please add such kind of information briefly.
Thanks for the comment. We added information in the abstract with the specific number of participants and their age range. The method is described with additional detail in the latter half of the abstract.
- Abstract, “The detection of psychological stress using … difficult to reliably detect.” is not necessary. It is better to be deleted or moved to Introduction section.
Thanks for the comment. This statement frames the purpose of the study, so the authors believe it should be left in.
- Fig. 1, duplicate “R-S interval”. Please delete one.
Thank you for catching this! It has been fixed.
- P2L70, Section ??, please confirm and revise.
Thanks for catching this! It has been fixed.
- Why two datasets were used?
Two data sets were not used. One data set was used, and two different models were used to detect stress; one model included features from small peaks within the ECG signal, and the other did not.
- How to annotate the “un-stressed” and “stressed”? What is the ground-truth for your annotation?
The ground truth annotation was added based on whether the video of the spider was playing or not. While this is not a direct measurement of stressed state, it is a proxy ground truth that we leveraged in the study. This limitation is acknowledged in the text of the paper.
- Section 2.2.1 and others, please tabulate such contexts.
Thank you for this suggestion. We have added Table 1 with the explanation of the HRV features.
- Fig. 2, Incomplete keywords
Thanks for catching this! The figures have been fixed.
- Fig. 2, top, middle, bottom -> top left, top right, bottom
Thanks for the suggestion, this change has been made.
- Tables, the title should go to the top of a table.
Thanks, this change has been made.
- Using small peaks is a good idea. However, detection of small peaks is not easy and reliable enough usually. Please discuss this issue and the impact on your classification performance.
This is a good point. Section 3.5 addresses this and shows the capability of an automated peak detection algorithm to detect the smaller peaks within the waveform. We address the concerns for our peak detection algorithm in the discussion.
- Please enhance the psychophysiological background and principle for including small peaks as new features.
Page 10, lines 297-317 add to the psychophysiological background and rationale for including the small peaks.
- The overall scientific writing is not satisfactory enough. It reads more like a student's lab report than a journal article. Thorough revision is necessary before resubmission.
Without specific examples of what makes this read like a student’s lab report, it is impossible to fix the error. Please articulate what elements need to be changed to improve the scientific writing. The authors believe that this paper has been written in a manner to enable understanding for people from a variety of different backgrounds, encouraging interdisciplinary collaboration. Please provide additional context to explain this comment.
Reviewer 2 Report
The article is interesting and in my opinion valuable. However, it still needs some improvements. I have the following comments:
1. The description of the methodology needs to be expanded. The addition of illustrations would be valuable. For example, diagrams, and an ECG waveform with marked features.
2. Text in figure 2 is cut off. The authors should fix it.
3. Figure captions should be short and detailed explanations should be in the text.
4. Many acronyms need to be explained. For example, the characteristics in Table 1. In addition, it would be useful to describe their interpretation.
5. There is a lack of explanation as to why the authors used these classifiers and not others. Furthermore, the authors did not propose the use of artificial neural networks, which is a modern approach. I know that dataset has few patients, but techniques such as transfer learning or few shot learning could be used. In my opinion, the study should be complemented by these methods.
6. There is no description of the technologies and tools used.
7. The authors did not compare their results to other research performed on the same dataset. This makes it impossible to clearly determine the effectiveness of the method.
8. It would be valuable to include a link to the source code (GitHub / GitLab) so that the experiment can be repeated.
Author Response
Reviewer 2:
The article is interesting and in my opinion valuable. However, it still needs some improvements. I have the following comments:
- The description of the methodology needs to be expanded. The addition of illustrations would be valuable. For example, diagrams, and an ECG waveform with marked features.
We have added detail to the methodology. Figure 1 includes an ECG waveform with features marked.
- Text in figure 2 is cut off. The authors should fix it.
Thanks for catching this! The figure has been fixed.
- Figure captions should be short and detailed explanations should be in the text.
Thank you for this comment. We have shortened the lengthy figure captions and incorporated those comments into the main text.
- Many acronyms need to be explained. For example, the characteristics in Table 1. In addition, it would be useful to describe their interpretation.
Section 2.2.1 Explanation of features now provides the explanation of each feature in Table 1.
- There is a lack of explanation as to why the authors used these classifiers and not others. Furthermore, the authors did not propose the use of artificial neural networks, which is a modern approach. I know that dataset has few patients, but techniques such as transfer learning or few shot learning could be used. In my opinion, the study should be complemented by these methods.
In this study, we aimed to perform classification with the simplest, most effective model, so this led us to use the methods covered in this paper. We agree that more modern approaches would complement the study, and we will plan to add this in future work, with a larger data set more suitable to that type of investigation.
- There is no description of the technologies and tools used.
The technologies and tools used to collect the data are described in Ihmig et al.’s cited publications. Given the focus on algorithm development in this paper, we leave the description of data collection to those publications.
- The authors did not compare their results to other research performed on the same dataset. This makes it impossible to clearly determine the effectiveness of the method.
This is a good point. The other publication that used this data set did not use the same method of ground truth, they used a subjective self-assessment of anxiety, so the model performance cannot be compared directly. The important comparison in this paper is between the novel feature model and traditional feature model to show the impact of adding in features derived from the small peaks in the ECG signal.
- It would be valuable to include a link to the source code (GitHub / GitLab) so that the experiment can be repeated.
This is a good suggestion. We would like to do this, but we have not put the code up on GitHub. Given the tight turnaround for this revision, it will not be possible to get this done this week, but we will plan to do it in the future. We should be doing this and will make sure to do so in the future!
Reviewer 3 Report
First of all, thanks for your new approach. I think this new approach is meaningful in that it significantly improves AIC. In addition, I think that the potential of the small peaks has been highlighted, thus increasing the possibility of sensor system development. This approach was good, but after I read this article, I have some questions and want some further explanation.
1. What is the low pass pole frequency of the Butterworth filter you used on line 113?
2. In the bottom matrix of Figure 2, I think it is difficult to judge the correlation only by shading between the traditional features and the novel features. I wonder if lower correlation coefficient is correct, so I want supplementary explanations (or numerical values).
3. I think the explanation about noise is insufficient. Data seems to be acquired in static environment. Please comment the ECG electrode position of the data and what types of noise could exist in your data. Finally, among the new features, which features do you think relatively robust to the noise?
4. How is the peak detection algorithm performance related to the thesis topic?
5. As the conclusion shows, only one of the three questions in lines 62-69 appears to have been resolved. I think the unresolved part needs further explanation (or the reason it doesn’t need to be resolved.)
6. Overall, I think the placement of tables, figures, and the body is weird, so it has poor readability. Also, the figure description is too long. It is recommended to shorten the figure description and add it to the body.
Again, thank you for writing this paper.
Author Response
Reviewer 3:
First of all, thanks for your new approach. I think this new approach is meaningful in that it significantly improves AIC. In addition, I think that the potential of the small peaks has been highlighted, thus increasing the possibility of sensor system development. This approach was good, but after I read this article, I have some questions and want some further explanation.
- What is the low pass pole frequency of the Butterworth filter you used on line 113?
Thanks for this comment, we have added the following text to address this: 5th order Butterworth filter with a normalized cutoff frequency of 0.3.
- In the bottom matrix of Figure 2, I think it is difficult to judge the correlation only by shading between the traditional features and the novel features. I wonder if lower correlation coefficient is correct, so I want supplementary explanations (or numerical values).
This is a good suggestion but adding numerical values to each box in the correlation matrices will be very difficult to read. Is this something that should be added in an appendix?
- I think the explanation about noise is insufficient. Data seems to be acquired in static environment. Please comment the ECG electrode position of the data and what types of noise could exist in your data. Finally, among the new features, which features do you think relatively robust to the noise?
Thanks for this comment. The original study states that the data set was collected with wearable sensors, so even though the participants were in a static environment, this type of sensor could have introduced noise. This is why we see some motion artifacts in the data. The S peak and T peak are both fairly robust to noise, this is described in more detail in Section 3.5.
- How is the peak detection algorithm performance related to the thesis topic?
Thanks for this comment. If the small peak features are shown to be useful in stress classification, it is important that we can actually collect those signals reliably. It is also essential that the peaks can be detected automatically, without time-consuming visual inspection. This is the relationship between these two areas for this work. We have added this reasoning to section 2.5.
- As the conclusion shows, only one of the three questions in lines 62-69 appears to have been resolved. I think the unresolved part needs further explanation (or the reason it doesn’t need to be resolved.)
Thank you for pointing this out! We have added to the conclusion to show our answers to the questions stated in the introduction.
- Overall, I think the placement of tables, figures, and the body is weird, so it has poor readability. Also, the figure description is too long. It is recommended to shorten the figure description and add it to the body.
Thanks for these comments. The figure descriptions have been shortened and that content has been better integrated into the text.
Again, thank you for writing this paper.
Thank you for the thoughtful review!
Round 2
Reviewer 1 Report
most of concerns raised by reviewers were treated but many careless mistakes still remain, very careful check and thorough revision are necessary before acceptance for publication.
Author Response
Thank you for the review. We have conducted a close check of the manuscript before resubmitting.
Reviewer 2 Report
1. The description of the methodology needs to be expanded. The addition of illustrations would be valuable. For example, diagrams, and an ECG waveform with marked features.
We have added detail to the methodology. Figure 1 includes an ECG waveform with features marked.
I may not have made it clear. I was referring to the real figures from dataset.
5. There is a lack of explanation as to why the authors used these classifiers and not others. Furthermore, the authors did not propose the use of artificial neural networks, which is a modern approach. I know that dataset has few patients, but techniques such as transfer learning or few shot learning could be used. In my opinion, the study should be complemented by these methods.
In this study, we aimed to perform classification with the simplest, most effective model, so this led us to use the methods covered in this paper. We agree that more modern approaches would complement the study, and we will plan to add this in future work, with a larger data set more suitable to that type of investigation.
As I said modern techniques can handle small datasets. In my opinion, in the current version, the paper is weak because it does not relate to modern techniques.
6. There is no description of the technologies and tools used.
The technologies and tools used to collect the data are described in Ihmig et al.’s cited publications. Given the focus on algorithm development in this paper, we leave the description of data collection to those publications.
I'm talking about technologies like programming language, libraries and hardware used to perform the research (e.g. algorithm implementation).
8. It would be valuable to include a link to the source code (GitHub / GitLab) so that the experiment can be repeated.
This is a good suggestion. We would like to do this, but we have not put the code up on GitHub. Given the tight turnaround for this revision, it will not be possible to get this done this week, but we will plan to do it in the future. We should be doing this and will make sure to do so in the future!
This should be done before the article is published, so that the link will be in the paper.
9. (new) References missing from text, question marks instead.
Author Response
1. Reviewer: The description of the methodology needs to be expanded. The addition of illustrations would be valuable. For example, diagrams, and an ECG waveform with marked features.
Author: We have added detail to the methodology. Figure 1 includes an ECG waveform with features marked.
Reviewer: I may not have made it clear. I was referring to the real figures from dataset.
Author: We have not included images from the data set, given that a single signal would not be representative of the data set as a whole. However, readers can access both the open-source data and now our code to inspect the data themselves.
5. Reviewer: There is a lack of explanation as to why the authors used these classifiers and not others. Furthermore, the authors did not propose the use of artificial neural networks, which is a modern approach. I know that dataset has few patients, but techniques such as transfer learning or few shot learning could be used. In my opinion, the study should be complemented by these methods.
Author: In this study, we aimed to perform classification with the simplest, most effective model, so this led us to use the methods covered in this paper. We agree that more modern approaches would complement the study, and we will plan to add this in future work, with a larger data set more suitable to that type of investigation.
Reviewer: As I said modern techniques can handle small datasets. In my opinion, in the current version, the paper is weak because it does not relate to modern techniques.
Author: We are committed to exploring other methods of data analysis in future investigations in this space, but we do not see the utility of adding these techniques to this work at this time. This first investigation shows that there is something of interest in the small peak features, we will leave it to future investigations with other data sets to investigate the predictive power that can be added with more "modern techniques".
6. Reviewer: There is no description of the technologies and tools used.
Author: The technologies and tools used to collect the data are described in Ihmig et al.’s cited publications. Given the focus on algorithm development in this paper, we leave the description of data collection to those publications.
Reviewer: I'm talking about technologies like programming language, libraries and hardware used to perform the research (e.g. algorithm implementation).
Author: The programming languages used were Python and R, and the primary package used to analyze the data is neurokit2. This is stated in the text.
8. Reviewer: It would be valuable to include a link to the source code (GitHub / GitLab) so that the experiment can be repeated.
Author: This is a good suggestion. We would like to do this, but we have not put the code up on GitHub. Given the tight turnaround for this revision, it will not be possible to get this done this week, but we will plan to do it in the future. We should be doing this and will make sure to do so in the future!
Reviewer: This should be done before the article is published, so that the link will be in the paper.
Author: This has been completed and the GitHub link is included in the data sharing portion of the paper.
9. Reviewer: (new) References missing from text, question marks instead.
Author: Thanks for catching this. The reference file was missing, this should be fixed now.
Reviewer 3 Report
The author revised the paper according to the reviewer's advice. Readability improved and it is clearer what you're describing. As the author answered, you don't have to worry about the second comment. Thank you for writing this paper.
Author Response
Thank you for your helpful review!